# UEA Digital Humans entry to the GENEA Challenge 2022

Jonathan Windle
University of East Anglia
Norwich, UK
j.windle@uea.ac.uk

David Greenwood
University of East Anglia
Norwich, UK
david.greenwood@uea.ac.uk

Sarah Taylor
Independent Researcher
Norwich, UK
sarah@saraht.co.uk

## ABSTRACT

This paper describes our entry to the GENEA (Generation and Evaluation of Non-verbal Behaviour for Embodied Agents) challenge 2022. The challenge aims to further the scientific knowledge using a large-scale, joint subjective evaluation of many gesture generation systems. We present two models to the challenge. A Bi-Directional LSTM for the full-body tier and a BDLSTM multi-decoder to produce body-section specific experts. We develop a loss function using both rotations and positions for training our models. We also introduce PASE+ features to the task of pose prediction, along with FastText word embeddings. Our models performed competitively regarding human likeness, and our multiple decoder system performed in the top two submissions for appropriateness of gesture.

## CCS CONCEPTS

• **Computing methodologies** → Intelligent agents; *Animation*.

## KEYWORDS

Speech-driven gesture generation, 3D Pose prediction, Neural Networks

**ACM Reference Format:**
Jonathan Windle, David Greenwood, and Sarah Taylor. 2022. UEA Digital Humans entry to the GENEA Challenge 2022. In *INTERNATIONAL CONFERENCE ON MULTIMODAL INTERACTION (ICMI '22), November 7–11, 2022, Bengaluru, India.* ACM, New York, NY, USA, 7 pages. https://doi.org/10.1145/3536221.3558065

## 1 INTRODUCTION & RELATED WORK

We participate in the 2022 GENEA challenge, submitting two systems. A Long Short-Term Memory (LSTM) baseline system was submitted to the full-body tier. An architecture with independent decoders for defined areas of the body was submitted to the upper-body tier. Each of these models are trained on the provided GENEA data [18] making use of the pre-trained PASE+ [13] speech audio encoder and pre-trained FastText [11] word encoder for multi-modal representations. Each system uses both audio and word embeddings to predict a sequence of 6D rotation [19] values for each body joint producing appropriate gesture animation.

There are many data-driven gesture generation techniques. Habibie et al. [6] utilise a Generative Adversarial Network (GAN) to model body, hand and face motion from audio. The generator in this model encodes audio speech using a 1D-convolutional neural network (CNN) and uses multiple decoders to predict motion. Pang et al. [12] also trained a GAN using an autoregressive generator. Word meaning and semantics have also been incorporated into gesture generation models using text-based features [9, 17]. Style control of synthesised motion was introduced by Alexanderson et al. with a flow-based model [1]. Taylor et al. also used a flow-based model, conditioned on speaking or listening [15].

Due to their strength in modelling sequential data, many speech-to-motion deep learning techniques are built upon bi-directional LSTMs [4, 7, 14]. LSTM-based models are a commonly used baseline in pose generation work [1, 8, 15]. We also train a bi-directional LSTM as our baseline model. Inspired by the multiple decoders used in Habibie et al. [6], we present a model that uses LSTMs to encode audio and text features and multiple LSTM-based decoders that model specific areas of the body. We divide the full body into 4 sections; head, upper body (including arms), hands and legs. We focus on extracting the most performance from a simple, easily accessible model and training procedure, and show novelty by using PASE+ [13] speech embeddings in conjunction with FastText [2] word embeddings, position and rotation in the loss function, and LSTM-based multi-head decoders for body parts.

Video examples and code can be found in the supplement at github.com/UEA-digital-human-group/GENEA22.

## 2 DATA PROCESSING

Our models used the supplied GENEA data [18] derived from the Talking With Hands dataset [10]. This data consists of high-quality 30fps mocap data in Biovision Hierarchical (BVH) format, with corresponding speech audio and text transcripts. Talking With Hands recorded dyadic conversations, however, the mocap and audio are separated by each speaker and in this challenge, treated independently. We use pre-trained models to encode the audio and text transcripts to descriptive feature vectors, suitable for gesture generation.

### 2.1 Motion Representation

A 3D pose is commonly represented by rotations and positions, in this work we utilise both representations but only predict rotations. We convert rotations to the 6D rotation representation presented by Zhou et al. [19]. These rotation representations have gained traction in 3D pose estimation recently [5, 16] due to Zhou et al. [19] finding these are more suitable for learning applications. Rotations can then be converted to 3D keypoint positions in world space.

As we are working with the BVH file format, there are two types of offset to consider. Global joint offsets and per-frame joint offsets.

In BVH format it is common to have a joint offset for each joint that represents each bone length. A per-frame joint offset is typically only present in the joint that represents world position, in the case of Talking With Hands format, the `body-world` joint. However, Talking With Hands is different in this regard as each joint has a per-frame offset too, possibly to account for bone-stretching in the data capture.

Talking With Hands contains multimodal data of multiple speakers and therefore different physical attributes. For each speaker identity, we observed a small difference in bone lengths between BVH files corresponding to the same speaker. This is likely due to the recording setup, however, the differences were minimal. For playback and BVH submission, we chose a single random BVH file for each speaker from the training dataset and used these values across all outputs for the respective speaker.

Regarding the per-frame offsets found in the Talking With Hands dataset, we observed the variance in these values to be low. Through visual inspection of the ground truth data, we observed that removing or keeping these values static throughout all frames did not impact visual performance. While our local playback of predicted motion was fine with the removed offsets, to ensure our BVH format was correctly formatted for the challenge, we added a static offset to each frame. This static offset was chosen from the same random BVH file per speaker as the joint offsets, but only the first frame offset was used and repeated across all frames in each BVH file. By keeping the bone lengths and per-frame offsets static, we believe this should allow the model to focus on representing the motion characteristics, rather than physical attributes.

## 2.2 Audio Representation

The most suitable audio representation for speech-motion synthesis is an open research question. One of the most common audio speech representations chosen in previous work is Mel Frequency Cepstral Coefficients (MFCCs) [1, 6, 15]. While this has provided impressive results, there is scope for more descriptive features. Through empirical evidence, we found that the problem-agnostic speech encoder (PASE+) [13] outperformed MFCCs. PASE+ adequately encodes an audio waveform to represent features required for 12 regression tasks. These 12 tasks include estimating MFCCs, FBANKs and other speech-related information including prosody and speech content. Therefore, MFCCs are implicitly encoded in these features as well as other useful speech-related features. PASE+ features are extracted before training. The PASE+ model expects audio waveforms to be sampled at 16KHz. Therefore the audio was downsampled using a band-sinc filtering method from 44.1KHz to 16KHz. We use the released, pre-trained PASE+ model to extract an audio feature embedding of size 768.

## 2.3 Text Representation

As a means to provide explicit word-based context to gesture, we include a text embedding to the model. We use the FastText word embedding described by Bojanowski et al. [2] using the pre-trained model released by Mikolov et al. [11]. This word embedding has been used in multi-modal gesture generation before [17] suggesting it is known to produce effective word embeddings for gesture generation. We extract each word embedding at a size of 300 per word

and its respective time frame within the context of the audio waveform. For each frame of motion, we include the word embedding of the word being spoken at the time of the frame. If no word is spoken at a given frame then a vector of zero values is passed. When a word is spoken across multiple frames, the vector is repeated for the appropriate number of frames.

## 3 METHOD

We introduce two models to the challenge. An LSTM-based baseline system to represent a reasonable performing, simple but effective method. This method was submitted to the full-body challenge. A second model involves the use of an LSTM encoder, followed by body-section-specific decoders. The encoder aims to represent the motion so that the decoders can each be specialists in predicting their respective body sections. This method was submitted to the upper-body challenge.

### 3.1 Data Presentation

Speaker identity is provided as a unique ID which we pass to an embedding layer. This layer contains a lookup table that stores a fixed vector embedding representative of the speaker. The layer contains trainable weights which means vector representations of speakers that move similarly should be close in vector space. This embedding acts as a style conditioning variable and produces motion that closely represents the style of the speaker ID provided. For this dataset, as there are only 17 different speaker identities, we found that the small embedding size of 2 is adequate to represent the different speaker styles.

We pre-process the speech audio and text transcripts as described in Section 2 before training. For both PASE+ and FastText models, these weights are frozen and not updated during training. Each data modality is then concatenated to a flat vector of size 1070 ready to be passed through the rest of the network.

### 3.2 LSTM Baseline

We first train a Bi-Directional LSTM baseline system. Figure 1 gives an end-to-end overview of the model. This model consists of 4 bi-directional layers, each with 1024 hidden units and a 40% dropout followed by a ReLU non-linearity layer and a fully connected layer. The output from the fully connected layer estimates the 6D rotations of each joint and the global position of the `body-world` joint.

### 3.3 Part-Specific Decoders

We provide a second architecture with part-specific expert decoders. An end-to-end view of this model is shown in Figure 2. Each decoder is responsible for a subset of joints representing the head, upper body (including arms), legs and hands. The encoder consists of 4 bi-directional layers, each with 768 hidden units and a 40% dropout followed by a ReLU non-linearity layer. This follows a similar architecture as our baseline and provides a good encoding of motion from our input. Each body section is predicted using a different decoder that each follows the same architecture. A decoder in the architecture consists of 2 bi-directional layers, each with 768 hidden units and a 40% dropout followed by a ReLU non-linearity layer and a fully connected layer. The output from each fully connected layer is the 6D rotations of representative joints.

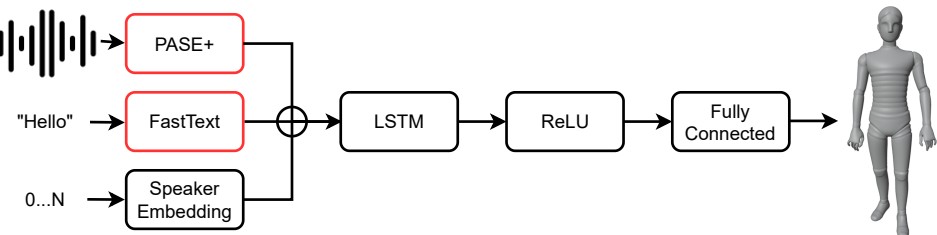

**Figure 1: Outline of our model used for full body speech-to-motion prediction. Our model takes as input speech audio, text transcript and a speaker encoding. Outputs are the joint rotation values. We use a pre-trained model for the audio and text inputs. Red box defines frozen weights.**

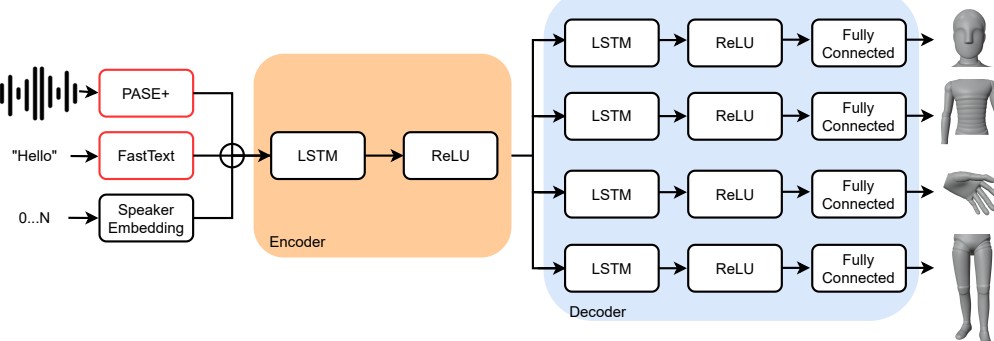

**Figure 2: Outline of the part-specific decoder model used for speech-to-motion prediction. Red box defines frozen weights**

The decoder responsible for the legs also predicts the `body-world` position as the leg movement should have the greatest impact on the global position of the speaker.

## 3.4 Training Procedure

We trained each model using the same procedure. The loss function contains multiple terms and weights. While we learn the 6D rotation values, we also include positions when computing the loss. We include an $L_2$ loss on the rotations, positions, acceleration and velocity of movement. By adding these terms, we qualitatively observed the motion became smoother and expanded the range of motion performed when compared to a rotation loss alone. Our final loss $L_c$ is computed as:

$$
\begin{aligned}
L_p &= \lambda_p L_2(y_p, \hat{y}_p) \\
L_v &= L_2(f'(y_p), f'(\hat{y}_p)) \\
L_a &= L_2(f''(y_p), f''(\hat{y}_p)) \\
L_r &= \lambda_r L_2(y_r, \hat{y}_r) \\
L_o &= \lambda_o L_2(y_o, \hat{y}_o) \\
L_c &= L_p + L_v + L_a + L_r + L_o
\end{aligned}
\tag{1}
$$

where $y_r$ and $\hat{y}_r$ are ground truth and predicted 6D rotations respectively, $y_p$ and $\hat{y}_p$ are the world positions derived from the 6D rotations and $y_o$ and $\hat{y}_o$ are the global offsets for the root joint. $f'$ and $f''$ are the first and second derivatives respectively. $L_p$ is

representative of positional distance, $L_v$ similarity in velocity, $L_a$ similarity in acceleration, $L_r$ is the similarity in 6D rotations and $L_o$ is how close the root offset is. $\lambda_p$ is the weighting of positions, $\lambda_r$ is the weighting of rotations and $\lambda_o$ is the weighting of offsets. These weights are applied to bring all terms into the same order of magnitude and increase the importance of some terms. $L_2$ represents the Mean Squared Error between the two sets of data. We used a small parameter search to find the optimal term weights. We observed that setting $\lambda_p = 0.1$, $\lambda_o = 0.01$ and $\lambda_r = 20$ produce the best motion.

The Adam optimiser is used during training with a learning rate of 0.0001 and a batch size of 256. Where hand motion is absent from the dataset, the hand motion is excluded during the loss calculation. This encourages the model to learn effective finger movements and avoid learning a static hand position. To balance training time and data samples, we split the motion into 30-frame chunks with the corresponding audio with a 25-frame overlap. Each model predicts a 30-frame sequence of motion, one frame at a time. We only train on the training data and leave out the validation data for model selection purposes. For the LSTM baseline, we train for 300 epochs and the part-specific decoder, 240 epochs determined by observed motion quality.

# 4 OBSERVATIONS

We observed two key issues. Rotations were sometimes predicted to unnatural values, particularly in the shoulders and arms. We also found that foot contact and natural leg movement was not always guaranteed.

## 4.1 Unconstrained Rotation

Although we found the inclusion of positions in our loss function to be beneficial, it introduced the issue of extreme rotations. If no weighting is applied to $L_p$ in Equation 1, this term dominates the loss and therefore caused unnatural rotations to be formed. This can be compared to solving inverse kinematics in that there are many solutions to form a particular pose position. We found that the model tended to produce impossible rotations, for example, rotations exceed a typical value range for a particular joint. Despite these physically impossible rotations, absolute positions of end-effectors relative to the over-rotated joint in world space appeared to be accurate.

Introducing a weight to constrain the positional influence allows a balance of valid rotation values and positive position influence. Despite the weight inclusion, there are still some issues regarding unnatural rotations. When viewing rendered sequences, we sometimes observed unnatural poses being formed. Figure 3a shows an example of a pose where the right shoulder has a rotation value outside of the typical range and caused an unnatural pose. We typically observed this issue would remain for several frames before recovering itself to return to a well-formed pose. Figure 3b shows a recovered pose of from the same sequence as Figure 3a. This issue is common in both proposed models, albeit slightly more prominent in the LSTM baseline. The motion predicted during these phases of over-rotation is still appropriate and gesturing still appears to be as correct to the speech as in other phases. We believe this could cause a negative effect when evaluating the human likeness of the predicted motion. However, we expect the appropriateness of gesture to be less affected.

## 4.2 Foot Contact

Our baseline LSTM model achieved some level of plausible leg movement and foot contact. However, we found our part-specific decoder model struggled to predict valid leg motion and foot contact. While some sequences of leg motion were realistic and appropriate, we often found the predicted leg motion involved large errors of foot contact where both feet are far from the ground.

Figure 4 shows an example of both legs raised unnaturally. While our results show that the part-specific decoders produce better arm, head and hand movement, this leg motion is very distracting and largely negates the good motion from the rest of the body. With this in mind, we chose to submit these model predictions only to the upper-body tier of the challenge. While the LSTM baseline predictions are submitted to the full-body tier.

# 5 RESULTS

Each model was evaluated in the user study in their respective tiers. The LSTM baseline is entered into the full-body tier with the ID **FSG** and the part-specific decoder model is entered into the

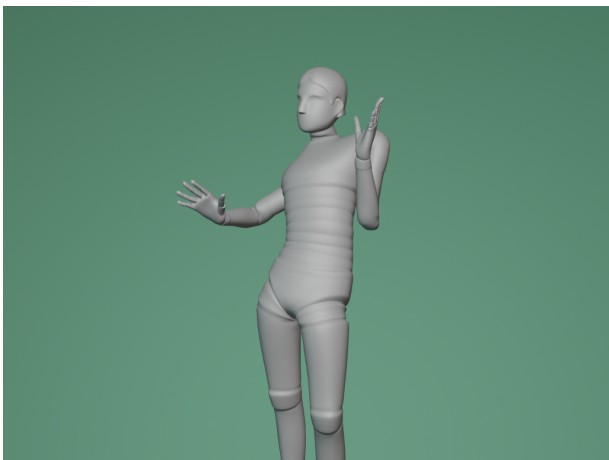

**(a) Effect of the shoulder joint exceeding its typical range.**

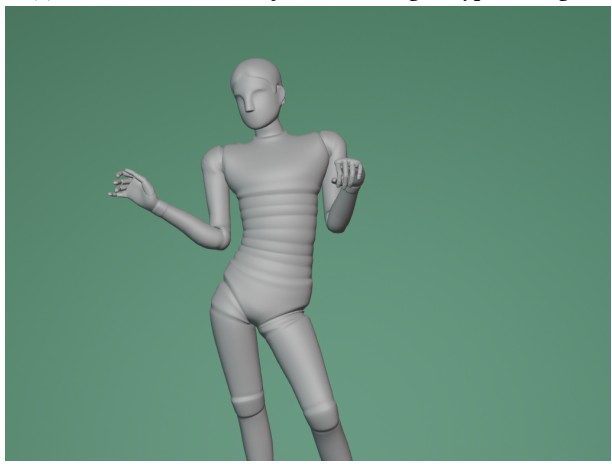

**(b) Recovered pose**

**Figure 3: An example of a sequence where a joint rotation exceeds a typical range of motion. In this case, the shoulder joint produces a rotation value which pushes the right arm back into an unnatural position. These unnatural poses typically resolve themselves after a while and we also show a pose from the same sequence once the rotation has recovered back to a normal range.**

upper-body tier with the ID **USM**. Table 1 provides results of the user-study from the main challenge paper [18].

## 5.1 Human-likeness

Both proposed models performed in the middle of the pack compared to all other submissions. This weakness of both models is likely due to the over-rotation issues described in Section 4.1.

While we can't compare the results of each model directly, we can compare each performance with their respective ground truth ratings. Although the upper-body median is only 3 higher, it is interesting to compare this against the median of the ground truth. The median rating of the LSTM baseline in the full-body study is 32 points lower than the ground truth. However, a lower median value

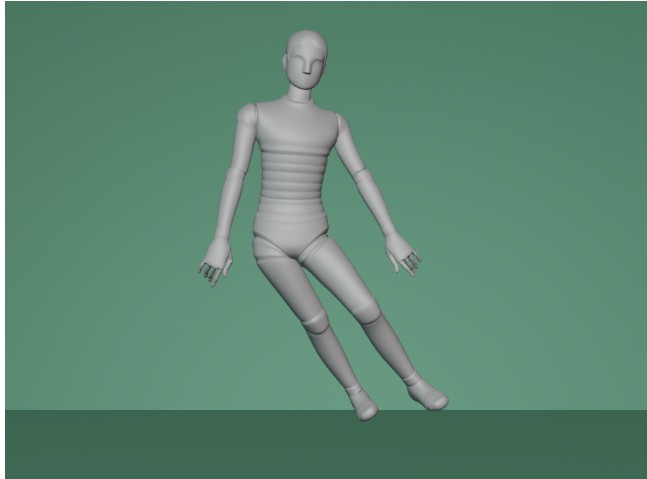

**Figure 4: Failure case for the part-specific decoder model incorrectly predicting leg motion. This shows a pose where both legs are visibly raised from the ground in an unnatural position for the legs.**

of the upper-body ground truth means that the gap between the part-specific decoder model and ground truth is 22. This suggests the part-specific decoder model may produce motion that is closer in human-likeness to the ground truth than the LSTM baseline.

Challenge organisers also included their baseline systems in the challenge. These use the IDs **FBT/UBT** for text-only baselines and **UBA** for the audio-only baselines. Figure 5 shows that in both challenge tiers our models are significantly better than all of the baselines.

## 5.2 Appropriateness

Where our models performed well was in the appropriateness of gesture to speech. Figure 6 visualises the distribution in responses from the appropriateness study. The full-body model remained in the middle of the pack, but can still be considered significantly more appropriate than random chance as the confidence interval does not overlap with the 0.5 value of random chance.

While we cannot draw a statistical significance against any other submissions, the fact that the upper-body submission went from the middle of the pack in human likeness to gaining the second highest appropriateness score in the submissions is promising.

It is difficult to derive the reason for this performance gain from the user study alone. However, we can speculate based on our visual observation. We observed gestures produced from this model would start at the expected time in relation to speech and the gesture intensity appeared to be an expected value too, particularly in the arm motion. The timing of beat gestures can be related to prosodic characteristics of speech [3]. We believe the observed accurate timing and intensity may come from the use of PASE+ features adequately encoding many speech features including prosody. As the arm-specific decoder only has to focus on predicting arms, we believe this decoder can more effectively use these features.

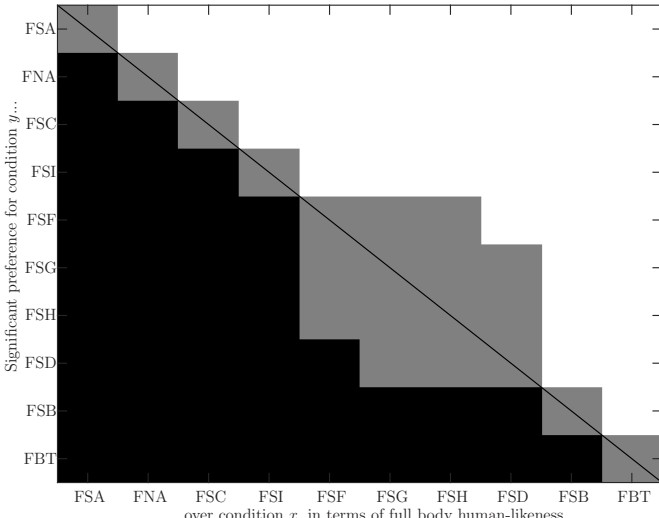

**(a) Full-body study**

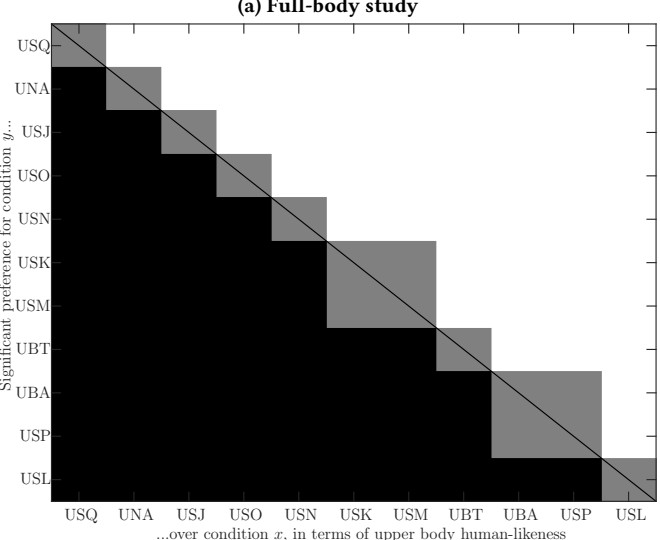

**(b) Upper-body study**

**Figure 5: Figure from main challenge paper [18]. Significance of pairwise differences between conditions. White means the condition listed on the $y$-axis rated significantly above the condition on the $x$-axis, black means the opposite ($y$ rated below $x$), and grey means no statistically significant difference at the level $\alpha = 0.05$ after Holm-Bonferroni correction.**

## 6 DISCUSSION

We have introduced two models to the challenge. While we are happy with their performance, there are still many things to consider going forward. A limiting factor in our predicted full-body motion is leg movement, particularly with the multiple decoder model. We believe this may be due to a weak correlation between speech and leg motion. Gestures are rarely made by legs alone and instead, the leg motion likely depends on the motion of the rest of the body. There appears to be a disparity between the leg movement

| | Human-likeness | | Appropriateness | | | |
|---|---|---|---|---|---|---|
| | | | Number of responses | | | Percent matched |
| ID | Median | Mean | Match. | Equal | Mismatch. | (splitting ties) |
| FNA | 70 ∈ [69, 71] | 66.7 ± 1.2 | 590 | 138 | 163 | 74.0 ∈ [70.9, 76.9] |
| FBT | 27.5 ∈ [25, 30] | 30.5 ± 1.4 | 278 | 362 | 250 | 51.6 ∈ [48.2, 55.0] |
| FSA | 71 ∈ [70, 73] | 68.1 ± 1.4 | 393 | 216 | 269 | 57.1 ∈ [53.7, 60.4] |
| FSB | 30 ∈ [28, 31] | 32.5 ± 1.5 | 397 | 163 | 330 | 53.8 ∈ [50.4, 57.1] |
| FSC | 53 ∈ [51, 55] | 52.3 ± 1.4 | 347 | 237 | 295 | 53.0 ∈ [49.5, 56.3] |
| FSD | 34 ∈ [32, 36] | 35.1 ± 1.4 | 329 | 256 | 302 | 51.5 ∈ [48.1, 54.9] |
| FSF | 38 ∈ [35, 40] | 38.3 ± 1.6 | 388 | 130 | 359 | 51.7 ∈ [48.2, 55.1] |
| **FSG** | 38 ∈ [35, 40] | 38.6 ± 1.6 | 406 | 184 | 319 | 54.8 ∈ [51.4, 58.1] |
| FSH | 36 ∈ [33, 38] | 36.6 ± 1.4 | 445 | 166 | 262 | 60.5 ∈ [57.1, 63.8] |
| FSI | 46 ∈ [45, 48] | 46.2 ± 1.3 | 403 | 178 | 312 | 55.1 ∈ [51.7, 58.4] |

**(a) Full Body Results**

| | Human-likeness | | Appropriateness | | | |
|---|---|---|---|---|---|---|
| | | | Number of responses | | | Percent matched |
| ID | Median | Mean | Match. | Equal | Mismatch. | (splitting ties) |
| UNA | 63 ∈ [61, 65] | 59.9 ± 1.3 | 691 | 107 | 189 | 75.4 ∈ [72.5, 78.1] |
| UBA | 33 ∈ [31, 34] | 34.6 ± 1.4 | 424 | 264 | 303 | 56.1 ∈ [52.9, 59.3] |
| UBT | 36 ∈ [34, 39] | 37.0 ± 1.4 | 341 | 367 | 287 | 52.7 ∈ [49.5, 55.9] |
| USJ | 53 ∈ [52, 55] | 53.6 ± 1.3 | 461 | 164 | 365 | 54.8 ∈ [51.6, 58.0] |
| USK | 41 ∈ [40, 44] | 41.5 ± 1.4 | 454 | 185 | 353 | 55.1 ∈ [51.9, 58.3] |
| USL | 22 ∈ [20, 25] | 27.2 ± 1.3 | 282 | 548 | 159 | 56.2 ∈ [53.0, 59.4] |
| **USM** | 41 ∈ [40, 42] | 41.9 ± 1.4 | 503 | 175 | 328 | 58.7 ∈ [55.5, 61.8] |
| USN | 44 ∈ [41, 45] | 44.2 ± 1.4 | 443 | 190 | 352 | 54.6 ∈ [51.4, 57.8] |
| USO | 48 ∈ [47, 50] | 47.3 ± 1.4 | 439 | 209 | 335 | 55.3 ∈ [52.1, 58.5] |
| USP | 29.5 ∈ [28, 31] | 32.4 ± 1.4 | 440 | 180 | 376 | 53.2 ∈ [50.0, 56.4] |
| USQ | 69 ∈ [68, 70] | 67.5 ± 1.2 | 504 | 182 | 310 | 59.7 ∈ [56.6, 62.9] |

**(b) Upper Body Results**

**Table 1: Table of results from main challenge paper [18]. Summary statistics of user-study ratings from all user studies, with confidence intervals at the level $\alpha$ = 0.05. "Percent matched" identifies how often participants preferred matched over mismatched motion in terms of appropriateness. Our model results are highlighted in pink . For Median, Mean, Match and Percent Matched columns, higher is better. For Mismatch, lower is better and for Equal, lower is preferable.**

and the rest of the body. Unfortunately without an entry for both models in both tiers, it is not possible to draw exact comparisons and improvements from one model to the other. We qualitatively observed evidence that the addition of independent decoders for separate parts of the body appears to work well and has been shown to work effectively in Habibie et al. [6]. Motion in the fingers, arms and head appear to improve over the LSTM baseline. Therefore it may be worth exploring separating the body into different sections in future. Decoding the legs with the core body may help with the disparity in leg movement.

Both models had lower scores for human likeness. We believe this is due to the occasional extreme rotation described in Section 4.1. In future work, it may be useful to include constraints on joints. For example, setting hard limits on how far a joint can rotate. These could either be learned from data or hand-crafted limits on a per-joint, per-speaker basis. Time and resources are limited. These models contain a large number of hyper-parameters that have a large impact on performance, particularly regarding

the weights defined in Equation 1. While we did perform a small parameter search, more performance could likely be gained from a more extensive parameter search.

## 7 CONCLUSION

We have presented our entries to the GENEA challenge 2022. We submitted an LSTM baseline to the full-body tier and a body-part-specific decoder architecture to the upper-body tier. Each of these models utilise the provided GENEA data and the pre-trained PASE+ [13] speech audio encoder and pre-trained FastText [11] word encoder. Each model performed reasonably in the middle of the pack of all submissions in the human likeness evaluation. The LSTM baseline performed in the middle of the pack in the appropriateness evaluation however, the part-specific decoder produced the second highest submission median score in the upper-body tier. We have discussed the weaknesses and strengths of these models and provided a discussion for future work.

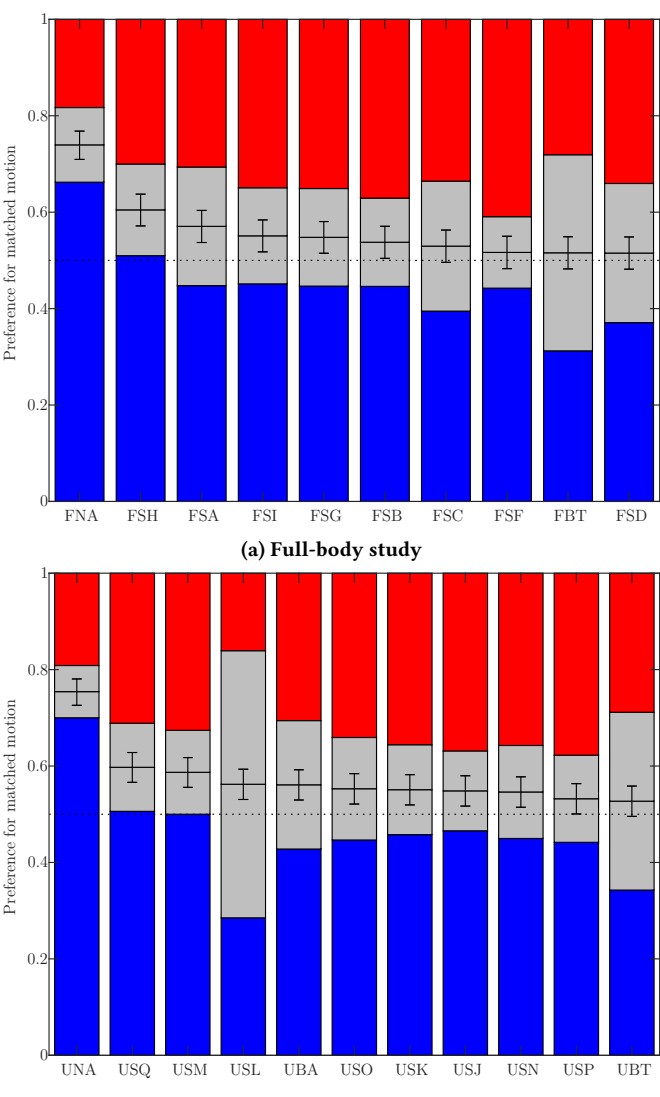

**(a) Full-body study**

**(b) Upper-body study**

**Figure 6: Figure from main challenge paper [18]. Bar plots visualising the response distribution in the appropriateness studies. The blue bar (bottom) represents responses where subjects preferred the matched motion, the light grey bar (middle) represents tied ("They are equal") responses, and the red bar (top) represents responses preferring mismatched motion, with the height of each bar being proportional to the fraction of responses in each category. The black horizontal line bisecting the light grey bar shows the proportion of matched responses after splitting ties, each with a 0.05 confidence interval. The dashed black line indicates chance-level performance. Conditions are ordered by descending preference for matched after splitting ties.**

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
