# OpenReview forum: "UEA Digital Humans entry to the GENEA Challenge 2022"
_ACM.org/ICMI/2022/Workshop/GENEA — GENEA Challenge & Workshop 2022 Mainproceeding_

### Official Review · Reviewer_bskq · 2022-08-08
**Anonymous entry to the GENEA Challenge 2022**

**Rating:** 6
**Confidence:** 5

**Review:**

This paper presents an approach using Bi-directional LSTM to generate human body gestures. The model receives audio, text transcription, and speaker ID as the inputs to produce human motion represented by joint rotation values. The authors applied the full-body framework for generating outputs submitted to the full-body tier, while the part-specific decoder was implemented in the upper-body tier.

Strength:
1. Overall, the paper is well-organized, allowing readers to catch up on the ideas quickly.
2. The paper presents several exciting ideas like encoding speaker ID and dividing a human body into sub-parts. However, I was expecting to see more in-depth discussions concerning the motivation and the effectiveness of those approaches to the generated motions.

Weakness:
1. Paper clarity: the authors should explain the proposed framework in more detail. In particular, the network architecture of the Speaker Embedding, Encoder, and Decoder. What is the motivation/benefit of adding the speaker embedding module to the network? Also, in the case of the part-specific decoder?

2. Technical concerns: How does the speaker embedding network update its weight values for distinguishing speaker IDs? Did the authors include the (classification?) loss in the total loss function? In terms of the part-specific decoders, it turns out that the four decoders (of the head, upper body, hands, and lower body) are independent networks. If that is the case, how did the authors guarantee the synchronization of the generated full body motion?

3. In-depth analysis: It would be more informative to have in-depth discussions about performance. Possibly, the author can conduct an ablation study (even with objective metrics) to discuss the effectiveness of the speaker embedding and the part-specific decoder.

---

### Official Review · Reviewer_d6NA · 2022-08-09
**Review of Anonymous entry to the GENEA Challenge 2022**

**Rating:** 7
**Confidence:** 3

**Review:**

Strong related works section and could even dive deeper into the long history of these models. (1) it hasn’t always been deep-learning only and (2) could describe the effects and purpose of adding word embeddings e.g. gestures carry semantic and communicative meeting relating to text.

Detailed description of data pre-processing which is crucial in this dataset.

Section 5.2: want more discussion of “appropriateness.” How was appropriateness evaluated and why does the model perform so well? What about the evaluation and model implementation helped it do well in this category? This seems like the punchline of the paper and both this and Human-likeness are only very lightly discussed. Please provide some interpretation or insight into why this model would be expected to perform well in this section.


**Nominate For A Reproducibility Award:**

Detailed description of models, background behind parameters chosen, and implementation. Assuming the code is released this could be nominated for this award.

---

### Official Review · Reviewer_PYyD · 2022-08-09
**A simple and effective framework**

**Rating:** 6
**Confidence:** 4

**Review:**

### Summary:
The paper presents a Bi-Directional LSTM framework for speech-to-motion prediction. The proposed method is simple and effective. An interesting part of this paper is the LSTM-based multi-head decoders for different body parts. According to the user study results, the proposed model performs well in terms of appropriateness, whereas the scores of human likeness are relatively lower due to the issue of extreme rotations.

### Strengths:
The technical details are presented in a straightforward manner. It's nice to see the analysis regarding the unconstrained rotations and foot contact, which definitely provides more insights into the prediction results.

### Weaknesses:
However, a weak part of the work lies in the fact that the multi-head decoders do not work well for the full body speech-to-motion prediction. Besides, I do have some concerns/comments listed below.

- The LSTM-based multi-head decoders are inspired by the multiple decoders used in Habibie et al. Therefore, the framework itself is not new and the novelty is somewhat incremental.

- Since the proposed system is a deterministic model, it would be better if the authors could provide some quantitative comparison with the ground truth, e.g., MSE.

- The data processing pipeline is not very clear. What are the feature sizes of the audio, text and speaker embeddings? How are these three modalities aligned?

- According to the paper (line 53), using the PASE+ [12] speech embedding is one of the contributions. In this sense, the comparisons with other commonly-used audio features, e.g., MFCC, would have been useful.

- Various loss functions have been applied (Eq 1). It would be helpful if the authors can provide more analysis on the effects of different terms.

### Minor:
- Is there any post-processing (e.g., smoothing etc.) that was applied to the predicted motion?
- In Eq 1, please indicate the meaning of f().
- Line 229-234: what’s the number of training epochs?

---

### Decision · Program_Chairs · 2022-08-11

**Decision:**

Accept (Main proceeding)

**Comment:**

All the reviewers were in favour of accepting this paper. The chairs agree to accept this paper. Please read the reviews carefully and revise the paper for the camera-ready version as follows:

1. Use your team name in the title and abstract.
2. Add missing details of the proposed method as three reviewers pointed out.
3. Try to reinforce the discussion part for the appropriateness results (commented by the reviewer d6NA). Your interpretation on the results or suggestion for future analysis would be helpful to other researchers.
4. Consider adding ablation studies if you have time to do.